# Analysis of Using the Parabolic Antenna as the Passive Calibrator for P-Band Spaceborne SAR Radiometric Calibration

**Shaoyan Du** [1,2], **Jun Hong** [1], **Yu Wang** [1,*], **Tian Qiu** [1,2], **Kaichu Xing** [1,2] **and Jianjun Huang** [3,4]

[1] National Key Laboratory of Science and Technology on Microwave Imaging, Aerospace Information Research Institute, Chinese Academy of Sciences, Beijing 100094, China; dushaoyan17@mails.ucas.ac.cn (S.D.); jhong@mail.ie.ac.cn (J.H.); qiutian19@mails.ucas.ac.cn (T.Q.); xingkaichu19@mails.ucas.ac.cn (K.X.)

[2] School of Electronic, Electrical and Communication Engineering, University of Chinese Academy of Sciences, Beijing 100049, China

[3] School of Electronics and Information Engineering, Beihang University, Beijing 100049, China; huangjianjun@buaa.edu.cn

[4] Beijing Institute of Remote Sensing Information, Beijing 100049, China

[*] Correspondence: wangyu@mail.ie.ac.cn

**Abstract:** The P-band spaceborne SAR system faces the problem of a lack of available reference targets for radiometric calibration. Traditional artificial passive calibrators such as trihedral corner reflectors may be hard to be adopted to P-band SAR calibration because of weight, size, manufacture, and installation cost. The parabolic antenna is considered as a potential calibrator due to its stable scattering characteristics and large RCS. One of the criteria of the calibrator is that the beamwidth of the azimuth RCS pattern needs to be much larger than the SAR antenna beamwidth, which is convenient for alignment and obtaining a constant RCS value. It is generally believed that the parabolic antenna can only be used when it is fully aligned to the SAR sensor. However, the P-band spaceborne SAR system usually has a wide beamwidth because of the long wavelength. In contrast, the azimuth RCS pattern of the parabolic antenna has a narrow beamwidth, which will cause inconstant RCS of the calibrator within the synthetic aperture period, thereby introducing an additional calibration error. Due to the imbalance between the SAR antenna beamwidth and the RCS pattern beamwidth, the criterion of using the parabolic antenna as a calibrator may need further discussion. This article analyzes the influence of the RCS pattern of the parabolic antenna on the radiometric calibration and establishes the quantitative error model. Based on the error model, the error analysis is carried out for the two cases of alignment and misalignment. The theoretical analysis shows that the calibration accuracy will decrease if without additional error compensation despite a fully aligned parabolic antenna. Under the case of misalignment, the parabolic antenna can also be used for calibration, even when introducing smaller errors. To verify this conclusion, the RCS pattern of a parabolic antenna is used for the P-band imaging simulation and calibration simulation. The results show that the narrow RCS pattern will introduce a non-negligible error in the calibration when fully aligned. The parabolic antenna can also assist in calibration in the case of misalignment. In addition, the calibration experiment shows that, after compensation with the error model, calibration accuracy has been effectively improved.

**Keywords:** P-band SAR; radiometric calibration; targets of opportunity

## 1. Introduction

Synthetic aperture radar (SAR) is widely used in military and civilian applications. SAR technology has been used in frequency at X-band, S-band, C-band, and L-band to achieve different tasks [1–4]. With the development of SAR technology, future spaceborne SAR missions will work in the P-band, such as BIOMASS. The P-band spaceborne SAR system can be used for global forest biomass and height acquiring, climate change research,

and soil moisture inversion, which is of great significance for the macroscopic observation of Earth and ecological protection [5–7]. Radiometric calibration is the prerequisite for all the quantitative applications mentioned above [8]. Due to the low frequency and long wavelength, the P-band SAR system will bring some challenges to radiometric calibration [9–13].

Radiometric calibration relies on the reference target with known RCS, such as the trihedral corner reflector (TCR) [14]. There is a criterion on the calibrator that the calibrator needs to have a larger RCS to ensure the sufficient signal-to-clutter ratio (SCR) [15]. The size of the TCR at different frequencies is also different. For example, when the SAR system works at X-band, a 1 m length TCR can meet the calibration requirement, while the L-band may need a 5 m length TCR. Due to the long wavelength, the P-band SAR system may require a TCR of more than 10 m. A 10m TCR needs at least 150 m$^2$ of materials and a volume of 500 m$^3$, which will bring difficulties and expensive costs to manufacturing and installation. In addition, the large-size TCR does not have a mature servo system, and considering the weight and size, the pointing angle is also difficult to deploy and control. Therefore, it is necessary to find an alternative artificial target as the calibrator.

The parabolic antenna may be an option as the potential reference target for calibration [16,17]. Due to the dish-shaped structure of the parabolic antenna, the volume and area of the parabolic antenna will be smaller than the TCR of the same size. Considering the mature manufacturing process and servo system of the parabolic antenna, using the parabolic antenna as the calibrator can reduce the cost and improve the feasibility. The idea of using the parabolic antenna as the calibrator is not new. European Space Agency (ESA) used a 4.7 m parabolic antenna for calibration in 1999–2000, and the experiment was conducted again in 2001 [18,19]. On the 12th Human Space Flight Memorial Day, ERS-2 was used to observe the 12 m and 15 m parabolic antennas in Spain [20]. The above experiments are performed in C-band but still verify the idea of using the parabolic antenna for image quality evaluation and calibration. Some research shows that this kind of target with large RCS can meet the RCS requirement of P-band spaceborne SAR radiometric calibration [21], but whether the parabolic antenna can be used for P-band calibration still needs further discussion.

The other criterion on the calibrator is that the beamwidth of the calibrator needs to be much larger than the beamwidth of the SAR antenna, which can facilitate the alignment and obtain a constant RCS value. In the calibration process, the RCS of the calibrator is considered as a constant. If the beamwidth of the calibrator's RCS pattern is close to being even narrower than SAR antenna beamwidth, the nominal RCS value of the calibrator may be inconstant during the calibration process, which will introduce uncertainty into the calibration. It is generally believed that the parabolic antenna can only be used as a calibration reference target when fully aligned with the SAR sensor. Unfortunately, the beamwidth of the parabolic antenna is usually very narrow, while the beamwidth of the P-band spaceborne SAR will be wide because of the long wavelength. Taking BIOMASS as an example, the diameter of the SAR antenna is 12 m, and the beamwidth is nearly 4°. Even if the pointing accuracy of the SAR antenna is good enough, due to the mismatch between the narrow beamwidth of the parabolic antenna and the wide beamwidth of the SAR antenna, the RCS of the parabolic antenna cannot be regarded as a constant, which will introduce an additional error [22–25]. This means that the parabolic antenna cannot be directly used for calibration without error compensation, even in the case of alignment. Therefore, a quantitative error model is necessary to use a parabolic antenna as a potential calibrator.

Parabolic antennas are widespread on the ground, and there are many large-size targets. The large parabolic antenna may still meet the SCR requirement in the case of misalignment, and there is a possibility to use it for calibration. If it works, more targets of opportunity could be found, which is meaningful for increasing calibration frequency. Therefore, the conditions of using the parabolic antenna as a calibrator also need to be further studied in the case of misalignment.

The purpose of this paper is to further demonstrate in detail the possibility of using a parabolic antenna as a reference target for P-band radiometric calibration to help solve the difficulty of lacking available passive calibrators. This paper first discusses the error introduced by the RCS pattern under a wide SAR beamwidth and establishes a quantitative model. Based on the error model, the conditions of using the parabolic antenna are reconsidered. Unlike the traditional understanding that it can only be used when fully aligned, the misalignment case is also discussed. P-band imaging and calibration simulation are performed with the RCS pattern of a typical parabolic antenna. The experiments first verify the error model to ensure that the error model can be used for compensation in the calibration process to improve the calibration accuracy. Secondly, the experiments indicate that the parabolic antenna can be used for calibration when not aligned with SAR. This conclusion expands the conditions of using the parabolic antenna.

Section 2 analyzes the scattering characteristics of the parabolic antenna, which is the basis for the possibility analysis of using it as a calibrator. Section 3 discusses the influence of the RCS pattern on the calibration, establishes the quantitative error model, and theoretically analyzes the conditions of using the parabolic antenna. The experiments are performed in Section 4. The conclusion is summarized in Section 5.

## 2. The Scattering Characteristic of Parabolic Antenna

In this section, the scattering characteristic of the parabolic antenna is simulated with FEKO, an electromagnetic simulation software that can calculate the RCS of the target. Due to the large size of the parabolic antenna, some electromagnetic methods such as Method of Moments (MoM) and Finite Difference Time Domain (FDTD) will face the problems of large computational complexity and high hardware requirements. Therefore, this paper chooses the high-frequency method Ray Launching-Geometrical Optics (RL-GO) to simulate the RCS characteristic of the parabolic antenna. To verify the correctness of RL-GO, the RCS pattern of a 9 m length TCR at the P-band is simulated. The geometric model and simulation results are shown in Figure 1. The simulated peak RCS value is 49.37 dBsm. The theoretical peak value of the TCR can be evaluated by the following:

$$RCS_{TCR} = \frac{4\pi b^4}{3\lambda^2} \tag{1}$$

where $b$ is the inner-leg length of the TCR, and $\lambda$ is the wavelength. The theoretical peak value of the TCR calculated with this formula is 49.44 dBsm. The simulated result is consistent with the theoretical result, which can verify the simulation method RL-GO.

The scattering characteristics of the target will be affected by azimuth angle and frequency, which means that the parabolic antenna has an angle-dependent and frequency-dependent RCS pattern. Therefore, the following will analyze the scattering characteristics in the azimuth RCS pattern and peak RCS over range bandwidth, respectively.

### 2.1. Azimuth RCS Pattern

A parabolic antenna with a diameter of 7.3 m is applied for simulation, and its geometric model is shown in Figure 2. A mental disk with a diameter of 0.4 m is placed in the focal area. The plane wave with 435 MHz illuminates at different azimuth angles, the elevation angle is fixed at 45°, and the azimuth angle ranges from −90° to 90°. RL-GO is used to simulate the RCS pattern of the parabolic antenna, and the simulated azimuth RCS pattern is shown in Figure 3.

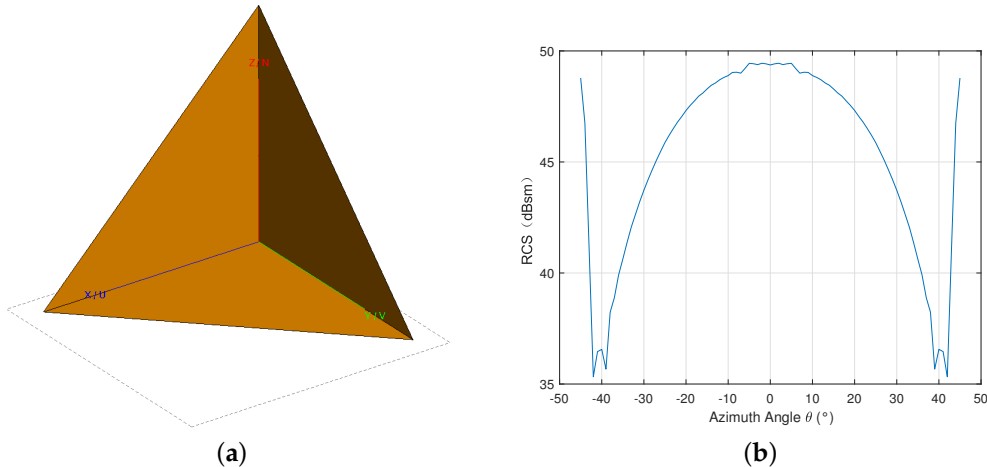

**Figure 1.** The scattering characteristic of the 9 m length TCR at P-band simulated with RL-GO. (**a**) Geometric Model of The 9 m length TCR; (**b**) RCS Pattern of The 9 m length TCR.

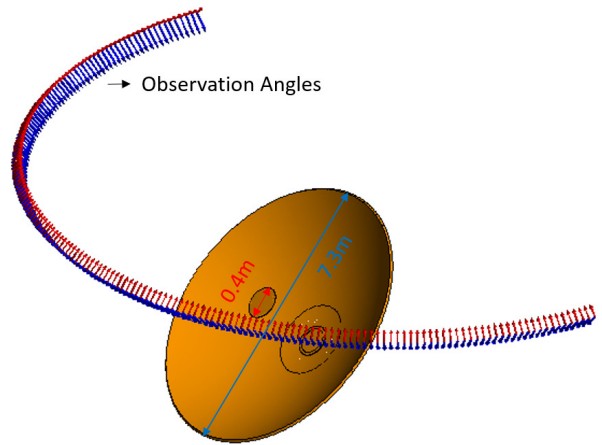

**Figure 2.** The geometric model of the parabolic antenna.

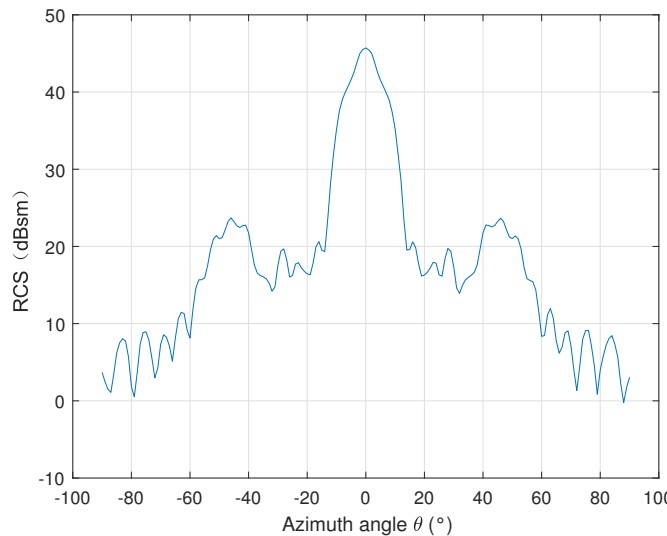

**Figure 3.** The RCS pattern of the parabolic antenna.

The peak RCS value of the parabolic antenna is 45.71 dBsm, and the 3 dB-beamwidth is 8°. Compared with the TCR, the beamwidth of the parabolic antenna is narrow. The

feed area of the parabolic antenna will affect the beamwidth of the RCS pattern. The larger the area, the wider the beamwidth. In this paper, a metal disc with a diameter of 0.4 m is placed in the focal area of the parabolic antenna. It is difficult for a real parabolic antenna to implement such a modification unless it is a fully cooperative target. Therefore, the RCS pattern of a common parabolic antenna may have a 3 dB beamwidth narrower than 8°.

The performance of the calibrator will affect the calibration result so that the calibrator has some criteria. One of the criteria is that the signal-to-clutter ratio (SCR) of the point target needs to be greater than 25 dB in order to ensure that the calibrator is visible in the scene, which can be expressed as the following formula:

$$SCR = \sigma_T - \sigma_C - 10\log_{10} p_a p_r \tag{2}$$

where $\sigma_T$ is the RCS of the calibrator, $\sigma_C$ is the normalized radar cross-section (NRCS) of the background clutter, and $p_a$ and $p_r$ are the resolution in azimuth and range. For the P-band spaceborne SAR system, the RCS requirement may be more stringent due to the lower resolution. It is assumed that the SCR of the calibrator must be greater than 30 dB. In the P-band, the NRCS of the desert is about −40 dB, the NRCS of the cultivated land is between −38 dB and −30 dB, and the NRCS of the forest is about −23 dB. Without loss of generality, assuming the background NRCS as $\sigma_C = -25$ dB, the azimuth resolution as $p_a$ = 25 m, and the range resolution as $p_r$ = 30 m, it can be obtained that when the RCS of the calibrator is greater than 40dB it may be used as a reference target for P-band calibration. According to this criterion, we can find that the RCS pattern of the parabolic antenna shown in Figure 3 has an azimuth angle range of 20° where the RCS is greater than 40 dB, which is much wider than the 3 dB beamwidth.

Another criterion of the calibrator is that the beamwidth of the calibrator needs to be greater than the beamwidth of the SAR antenna. The wider the beamwidth of the calibrator, the easier the SAR antenna aligns to the calibrator. At the same time, the measured RCS of the calibrator can be closer to the nominal RCS value so that the less error. The criterion can be formed mathematically as follows:

$$BW_{cal} > \phi_a + \Delta\phi_y = BW_{SAR} \tag{3}$$

where $BW_{SAR}$ is the beamwidth of the SAR composed of the SAR antenna beamwidth $\phi_a$ and the yaw attitude stability of the antenna $\Delta\phi_y$. $BW_{cal}$ is the beamwidth of the calibrator, which should be larger than the $BW_{SAR}$. For high-frequency spaceborne SAR systems, this is not difficult to meet because the beamwidth of the TCR is about 40°, and the beamwidth of the SAR antenna is usually narrower than 1°. However, when the frequency is reduced to the P-band, the SAR antenna beamwidth will expand. The following formula can estimate the SAR antenna beamwidth.

$$\theta_{3\text{dB}} \approx 1.22 \frac{\lambda}{D_{\text{ref}}} \tag{4}$$

BIOMASS uses an array-fed reflector antenna with a diameter of 12 m and works at 435 MHz. According to the Equation (4), the SAR antenna beamwidth is about 4.11°. The beamwidth of the parabolic antenna is not much different from the beamwidth of the SAR antenna, which may make the use of the parabolic antenna very difficult. Therefore, whether the parabolic antenna can be used for calibration needs further discussion.

It is generally believed that the parabolic antenna can only be used for calibration when fully aligned with the SAR. Based on the above analysis, we need to consider two issues. First, although the beamwidth of the parabolic antenna is narrow, the angle range that meets the RCS requirement of the calibration is extensive. When the calibrator is not aligned with SAR, whether it can be used for calibration needs to be analyzed. Second, the beamwidth of the parabolic antenna does not have the advantage over the beamwidth of the SAR antenna. The RCS of the target changes drastically during the synthetic period and cannot be regarded as a constant. Although the parabolic antenna is fully aligned with

the SAR, it will still affect the calibration accuracy. Therefore, a compensation method is necessary. These issues will be discussed in Section 3.

### 2.2. RCS over Range Bandwidth

The RCS of the calibrator is also affected by frequency. When the bandwidth of the SAR system is wide, the frequency-dependent effect may not be negligible because it will make the RCS inconstant within the bandwidth. Therefore, before using the parabolic antenna as a calibrator, we need to analyze the influence of the bandwidth on the scattering characteristics. The mission of the P-band spaceborne SAR system is macro observation, and high resolution is not the goal, so the bandwidth is relatively narrow. Still taking BIOMASS as an example, the bandwidth is 6 MHz. The RCS of the parabolic antenna over the bandwidth of 6 MHz is shown in Figure 4a.

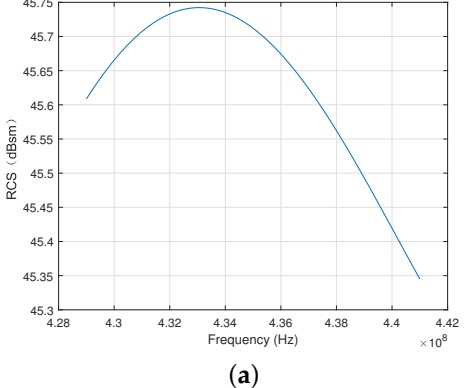 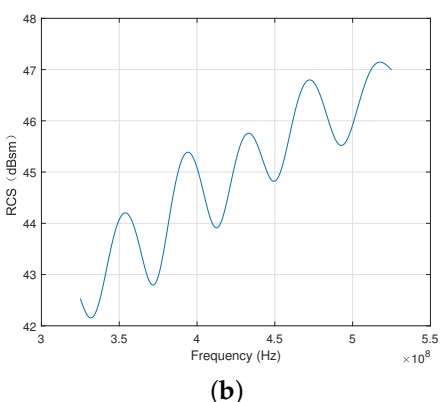

(**a**)    (**b**)

**Figure 4.** The range RCS pattern of the parabolic antenna. (**a**) Range RCS Pattern with Bandwidth = 6 MHz; (**b**) range RCS Pattern with Bandwidth = 100 MHz.

The result shows that the maximum difference of the RCS is less than 0.3 dBsm, so the influence is negligible. For further analysis, the bandwidth is widened to 100 MHz, as shown in Figure 4b. As the bandwidth increases, the RCS changes more drastically, the influence of frequency-dependent RCS must be considered. This paper assumes the RCS of the parabolic antenna over the range bandwidth is constant during the calibration process so that no further discussion will be provided.

## 3. The Conditions of Using the Parabolic Antenna

As mentioned in Section 2, the RCS pattern of the parabolic antenna varies drastically along the azimuth angle, and the SAR antenna beamwidth is wide. Hence, the influence of the azimuth RCS pattern on the calibration needs to be considered. This section will analyze the quantitative formula of the additional error introduced by the azimuth RCS pattern under the wide SAR beamwidth and discuss the conditions of using the parabolic antenna based on the error model.

### 3.1. The Influence of the Azimuth RCS Pattern

The calibrated SAR image can be directly related to the backscatter coefficient or RCS of the scene according to the radiometric offsets, described by the radiometric coefficient $K$, which is used to scale the pixel intensities to achieve the mapping from the pixel intensity to RCS [26,27]. In the practical calibration process, the nominal RCS value of the target is a single value, usually the peak value of the azimuth RCS pattern. Under this assumption, the integral of the pixel intensity, that is, the energy of the point target, is proportional to the RCS of the target [28,29]. The expression for the backscatter coefficient or RCS is derived from the complex SAR image using an integral approach. In this method, the gain $K$ is estimated by integrating the pixel power over an area containing the calibration target and subtracting the contribution from the clutter and noise. The contribution from noise

and clutter will be subtracted. For the convenience of discussion, the influence of noise is ignored, and a convenient mathematical expression of a complex SAR image is formed as follows:

$$V(x,y) = \sqrt{K}S(x,y) \otimes h(x,y) \tag{5}$$

where $K$ is the gain imposed by the SAR on the backscatter measurement, and $h$ is the SAR system image blurring function, often referred to as the point-spread impulse response function. The $x$ coordinate can represent the direction of azimuth time or azimuth spatial position, and the $y$ coordinate represents the range direction. There is an assumption that the azimuth and range directions are independent. $S(x,y)$ is a function of spatial position $(x,y)$, which represents the known point target, and can be described by the following:

$$S(x,y) = \sqrt{\sigma}\delta(x)\delta(y) \tag{6}$$

where $\sigma$ is the known RCS of the point target. At this point, the nominal value of the known RCS is a constant. $\delta(x)$ and $\delta(y)$ are the Dirac function in azimuth direction $x$ and range direction $y$. The SAR image pixel values can be related to the RCS after square-law detection, which can be derived as follows:

$$P(x,y) = |V(x,y)|^2 = K\sigma|h'(x,y)|^2 \tag{7}$$

where the following is the case.

$$h' = \delta(x)\delta(y) \otimes h(x,y) \tag{8}$$

The impulse response function for the square-law detected image $h'$ is related to the sampling intervals and the partial coherence of the system, scene, and processor. The relation may not be developed, but there is a restriction that the integral of $h'$ over image region is unity. Therefore, when the integral area $A$ containing the point target response is sufficiently large, an approximate is valid.

$$\int\int_A |\delta(x,y) \otimes h(x,y)|dxdy \simeq 1 \tag{9}$$

Thus, the integrated point target pixels, that is, the energy of the point target, can be written as follows.

$$I = \int\int_A P(x,y)dxdy = K\sigma \tag{10}$$

The above derivation expresses the conventional understanding of the relation between the SAR image pixel intensity and the known RCS. If the beamwidth of the SAR antenna becomes wider or the requirements for calibration accuracy increase, it is not enough to treat the nominal value of the known RCS as a constant, which will introduce errors to the calibration.

Due to the independence of the azimuth direction and the range direction, without loss of generality, the received signal after range compression can be expressed as follows.

$$S_r = \sqrt{\sigma(x)} \cdot \delta(y) \tag{11}$$

The azimuth RCS pattern $\sigma(x)$ is the function of $x$. Due to the azimuth phase added by the Doppler effect, the azimuth signal can also be regarded as a chirp signal. The azimuth signal can be simplified as follows:

$$S_a(t) = \sqrt{\sigma(t)} \cdot e^{j\pi K_a t^2} \tag{12}$$

where $t$ is the azimuth time, $K_a$ is the Doppler frequency rate, and the RCS $\sigma(t)$ is converted from the azimuth angle domain to the azimuth time domain by the following:

$$\theta = \arctan\left(\frac{v \cdot t}{R}\right) \tag{13}$$

where $v$ is the speed of the SAR sensor, and $R$ is the slant range between the SAR sensor and the calibration target. After azimuth compression, the signal becomes the following:

$$
\begin{aligned}
S_a &= \sqrt{\sigma(t)} \cdot e^{j\pi K_a t^2} \otimes e^{-j\pi K_a t^2} \\
&= \int_{-\infty}^{\infty} \sqrt{\sigma(t)} \cdot e^{j\pi K_a \tau^2} e^{-j\pi K_a (\tau-t)^2} d\tau \\
&= e^{-j\pi K_a t^2} \cdot \int_{(\tau-\frac{T}{2})}^{(\tau+\frac{T}{2})} \sqrt{\sigma(\tau)} \cdot e^{j2\pi K_a t\tau} d\tau \\
&= e^{j\pi K_a t^2} \cdot \int_{-\frac{T}{2}}^{\frac{T}{2}} \sqrt{\sigma(\tau)} \cdot e^{j2\pi K_a t\tau} d\tau \\
&= \sqrt{K} \cdot \mathcal{F}\left[\sqrt{\sigma(\tau)}\right] \cdot e^{j\pi K_a t^2}
\end{aligned}
\tag{14}
$$

where $T$ is the synthetic aperture time. The above formula contains three parts: $\sqrt{K}$ is the processed gain, and $\mathcal{F}\left[\sqrt{\sigma(\tau)}\right]$ is the Fourier transform of the azimuth RCS pattern. It can be inferred that the Fourier transform of the azimuth RCS pattern is the envelope of the azimuth slice of the point target. The third term $e^{j\pi K_a t^2}$ is the additional phase, and the peak of this term near $t = 0$ is usually negligible in the main lobe so that the formula can be approximated as follows:

$$S_a \approx \sqrt{K} \cdot \mathcal{F}\left[\sqrt{\sigma(t)}\right] \tag{15}$$

At this point, the energy of the point target can be calculated by integrating the signal that after the azimuth compression in the synthetic aperture time, the detailed expression is as follows.

$$
\begin{aligned}
I_A &= \int_{-\frac{T}{2}}^{\frac{T}{2}} \left|\sqrt{K} \cdot \mathcal{F}\left[\sqrt{\sigma(t)}\right]\right|^2 dt \\
&= K \cdot \int_{-\frac{T}{2}}^{\frac{T}{2}} \left|\mathcal{F}\left[\sqrt{\sigma(t)}\right]\right|^2 dt
\end{aligned}
\tag{16}
$$

According to Parseval's theorem, the energy of the signal in the time domain is equal to the energy of its Fourier transform in the frequency domain such that Equation (16) can be further expressed as follows.

$$I_A = K \cdot \int_{-\frac{T}{2}}^{\frac{T}{2}} \sigma(t) dt \tag{17}$$

The difference from the traditional relation is that when the azimuth RCS pattern is taken into account, the energy of the point target is proportional to the integral of the azimuth RCS pattern in the azimuth time. This conclusion can also apply to the case where the nominal value of the known RCS is a constant. When the RCS of the target is a constant $\sigma$, according to Equation (17), the point target energy can be expressed as follows.

$$
\begin{aligned}
I &= K \cdot \int_{-\frac{T}{2}}^{\frac{T}{2}} \sigma dx \\
&= KT\sigma
\end{aligned}
\tag{18}
$$

By including the synthetic aperture period $T$ into the gain $K$, it is obtained in the same form as Equation (10):

$$I = K'\sigma \tag{19}$$

According to the modified relation, the error introduced by ignoring the azimuth RCS pattern can be estimated by calculating the energy difference between the two cases where the nominal RCS value is regarded as a constant, and another the azimuth RCS pattern is taken into account. The error can be expressed as follows:

$$\Delta = 10\log_{10}\left(\frac{I_r}{I_s}\right) \tag{20}$$

where $I_r$ is the energy of the calibration target corresponding to the azimuth RCS pattern, and $I_s$ is the energy of the calibrator corresponding to the single RCS value. Furthermore, the ideal energy $I_s$ can be expressed as follows.

$$I_s = K \cdot \int_{-\frac{T}{2}}^{\frac{T}{2}} \sigma(0)dt \tag{21}$$

Here is an assumption that the value of the azimuth pattern at the azimuth central moment is the ideal nominal RCS value. Substituting Equations (17) and (21) into the Equation (20), the final error expression can be obtained.

$$\Delta = 10\log_{10}\left(\frac{\int_{-\frac{T}{2}}^{\frac{T}{2}} \sigma(t)dt}{\int_{-\frac{T}{2}}^{\frac{T}{2}} \sigma(0)dt}\right) \tag{22}$$

### 3.2. The Error Analysis of the Parabolic Antenna

This part will discuss the conditions of using the parabolic antenna for calibration based on the error model. As mentioned in Section 2, two issues should be considered. First, when the parabolic antenna is fully aligned, what impact will the inconstant azimuth RCS pattern have on the calibration? Second, when the parabolic antenna is not aligned with the SAR, can it be used for calibration when the SCR requirement is still met? These questions could be discussed in two situations. When the parabolic antenna is aligned with the SAR, the azimuth RCS pattern can be approximated as a quadratic function model. When the parabolic antenna is not aligned, the azimuth RCS pattern can be abstracted to a linear function, and the following will discuss the two cases separately.

3.2.1. Linear Model

When the azimuth angle $\theta = -6°$, the azimuth RCS pattern of the parabolic antenna is approximately a linear model within the synthetic aperture period, as shown in Figure 5. According to the configuration of BIOMASS, the beamwidth of the SAR antenna is set to $4°$. The orange dotted line is the ideal nominal RCS value that keeps constant, and the blue line is the partial RCS pattern of the parabolic antenna.

Assuming that the azimuth RCS pattern obeys a linear function, we have the following.

$$\sigma_L = ax + b \tag{23}$$

By substituting the RCS pattern Equation (23) into Equation (22), we can obtain the error caused by the linear azimuth RCS pattern.

$$\begin{aligned}
\Delta_L &= 10\log_{10}\left(\frac{\int_{-\frac{T}{2}}^{\frac{T}{2}} ax + b\, dt}{\int_{-\frac{T}{2}}^{\frac{T}{2}} b\, dt}\right) \\
&= 10\log_{10}\left(\frac{\frac{1}{2}ax^2 + bx \, |_{-\frac{T}{2}}^{\frac{T}{2}}}{bx \, |_{-\frac{T}{2}}^{\frac{T}{2}}}\right) \\
&= 0
\end{aligned} \tag{24}$$

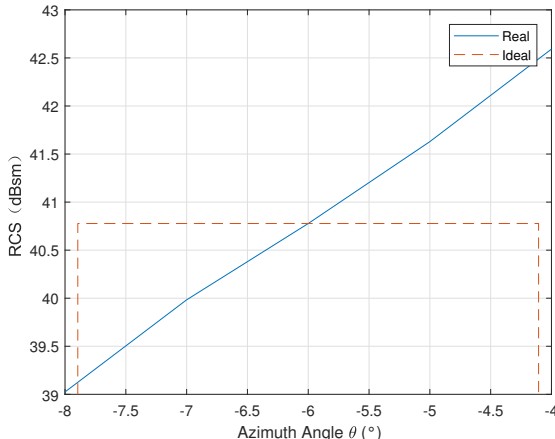

**Figure 5.** The partial azimuth RCS pattern in the case of misalignment ($\theta = -6°$).

In this case, the RCS pattern will not theoretically introduce errors, so the nominal RCS value of the calibrator can still be regarded as a constant. This conclusion indicates that if the parabolic antenna meets the RCS requirement and the azimuth pattern is approximately a linear function, it may also be used as a calibrator even in the case of misalignment. To a certain extent, this can make the conditions of using the parabolic antenna simpler. Of course, it is hard for the azimuth RCS pattern of the parabolic antenna to be a perfect linear function, and error compensation is also required at this point.

### 3.2.2. Quadratic Model

This section analyzes the azimuth RCS pattern corresponding to the alignment, and the RCS pattern of the parabolic antenna within the SAR antenna beamwidth is shown in Figure 6. Similarly, by assuming that the azimuth RCS pattern of this target obeys the following quadratic function, we have the following.

$$\sigma_Q = ax^2 + bx + c \tag{25}$$

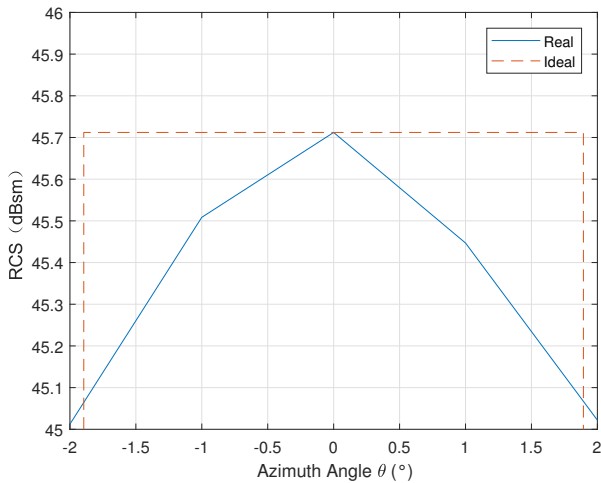

**Figure 6.** The partial azimuth RCS pattern in the case of alignment ($\theta = 0°$).

By substituting the quadratic RCS pattern Equation (25) into Equation (22), we can obtain the corresponding error:

$$\Delta_Q = 10\log_{10}\left( \frac{\int_{-\frac{T}{2}}^{\frac{T}{2}} ax^2 + bx + cdt}{\int_{-\frac{T}{2}}^{\frac{T}{2}} cdt} \right)$$

$$= 10\log_{10}\left( \frac{\frac{1}{3}ax^3 + \frac{1}{2}bx^2 + cx\big|_{-\frac{T}{2}}^{\frac{T}{2}}}{cx\big|_{-\frac{T}{2}}^{\frac{T}{2}}} \right)$$

$$= 10\log_{10}\left( \frac{\frac{aT^2}{12} + c}{c} \right) \tag{26}$$

The radiometric accuracy of the current spaceborne SAR system is typically below 1 dB, so the error should be controlled within a relatively small range. Assuming that the error caused by the RCS pattern is $\eta$ that is smaller than 1dB [30,31], the error $\Delta_L$ should be within this range.

$$10^{-\frac{\eta}{10}} < \frac{\frac{aT^2}{12} + c}{c} < 10^{\frac{\eta}{10}} \tag{27}$$

The above analysis shows that even if the parabolic antenna is perfectly aligned with the SAR, the calibrator with narrow beamwidth will introduce errors. If the parabolic antenna takes some modification, such as placing a metal disc in the focal area to expand the beamwidth, the additional error may be limited to an acceptable range. However, for most cases, error compensation is an easier method to guarantee calibration accuracy.

## 4. Experiments

In the previous sections, we have theoretically analyzed the criterion of using the parabolic antenna for calibration. When fully aligned, an error will be introduced due to the inconstant RCS, and additional compensation is required; when not aligned, if the RCS pattern is approximate to a linear model and meets the SCR requirement, the parabolic antenna can also be used as a potential calibrator, even when out of the 3 dB beamwidth. This section will verify the above conclusion by performing P-band radiometric calibration simulation with the azimuth RCS pattern.

*4.1. Error Model Verification*

4.1.1. Alignment

According to Section 3.2.2, an additional error will be introduced when the parabolic antenna is aligned with the SAR sensor. Imaging simulation based on the azimuth RCS pattern is performed first to verify the error model. The detailed SAR system parameters are shown in Table 1.

Define the pointing deviation as the angle that the antenna deviates from the central angle of the RCS pattern. Figure 7 shows the normalized RCS patterns with different pointing deviations. The experiments start with performing imaging simulation based on these azimuth RCS patterns, the partial results are shown in Figure 8, and the narrow beamwidth of the parabolic antenna does not affect point-like target formation.

**Table 1.** P-band SAR System Parameters.

| Parameter | Value |
|---|---|
| Frequency | 435 MHz |
| Bandwidth | 6 MHz |
| Pulse width | 40 μs |
| Altitude | 750 km |
| Velocity | 7.1 km/s |
| Az.Antenna Size | 12 m |
| Az.Antenna beamwidth | 4.11° |

After obtaining the SAR image of the calibrators, the point target energy can be extracted through the integral approach. The simulated data are shown in Table 2, where $\Delta_s = I_r - I_s$ means the error predicted by numerical simulation, $\Delta_t$ means the error predicted by theoretical error model Equation (22), and the difference $|\Delta_s - \Delta_t|$ is used to verify the accuracy of the proposed error model. If the difference is acceptable, the error model is valid. The point target energy with different pointing deviations, as shown in Figure 7, is simulated for further verification.

The result shows that the difference between the error predicted by numerical simulation and the error predicted by the theoretical formula is less than 0.01 dB, which verifies the error model. In addition, the results also indicate that as the SAR pointing deviation increases, the error changes are not uniform. When the pointing deviation is 3°, the error is only 0.1 dB. This is because the RCS pattern with a pointing deviation of 3° is close to the linear model. However, when the pointing deviation is 2° and 4°, the error rises to 0.3 dB again. Compared to the situation when the pointing deviation is 3°, the RCS pattern changes more drastically at other pointing deviations.

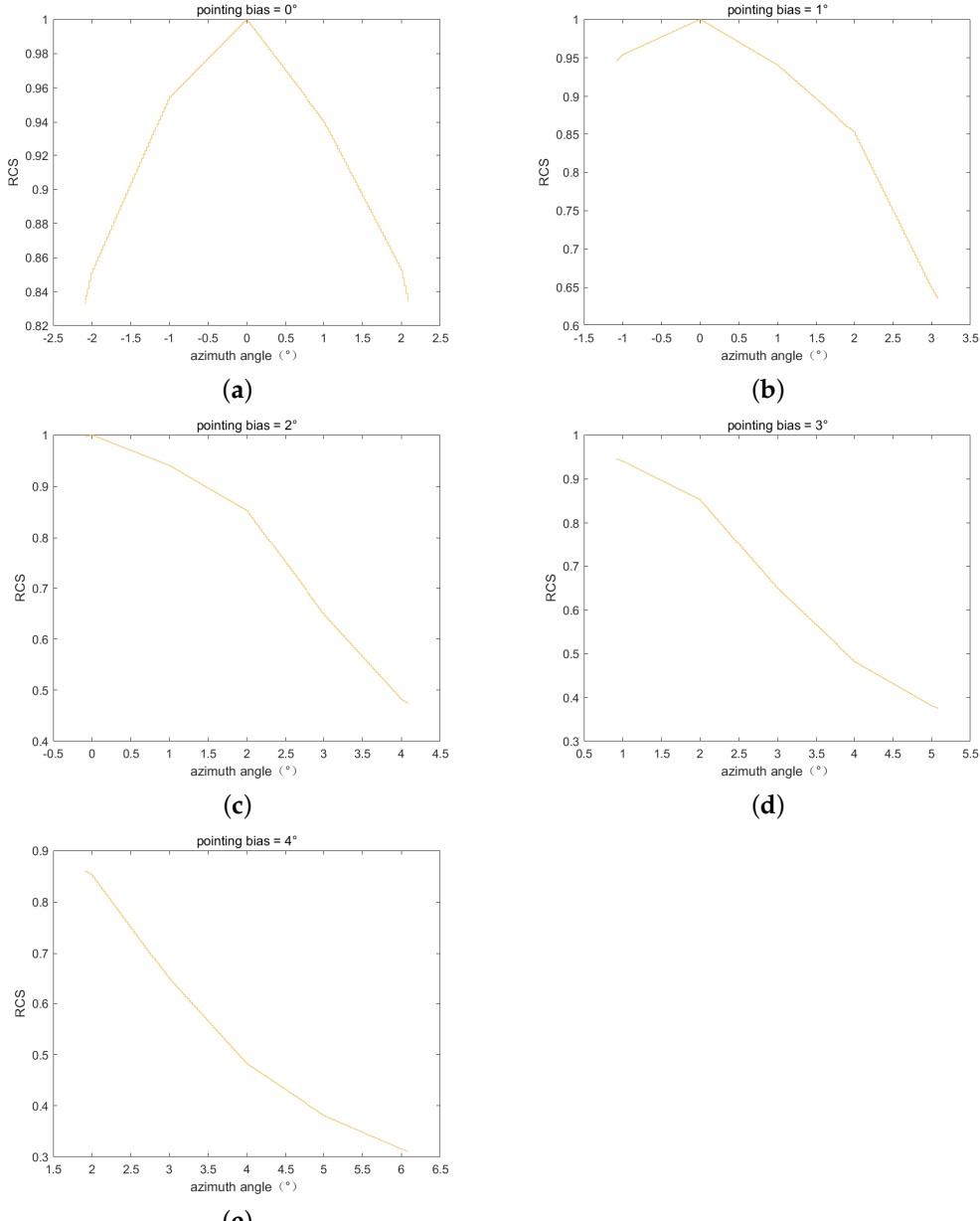

**Figure 7.** The normalized RCS pattern with different pointing deviations in the case of alignment. (**a**) Pointing deviation = 0°; (**b**) pointing deviation = 1°; (**c**) pointing deviation = 2°; (**d**) pointing deviation = 3°; (**e**) pointing deviation = 4°.

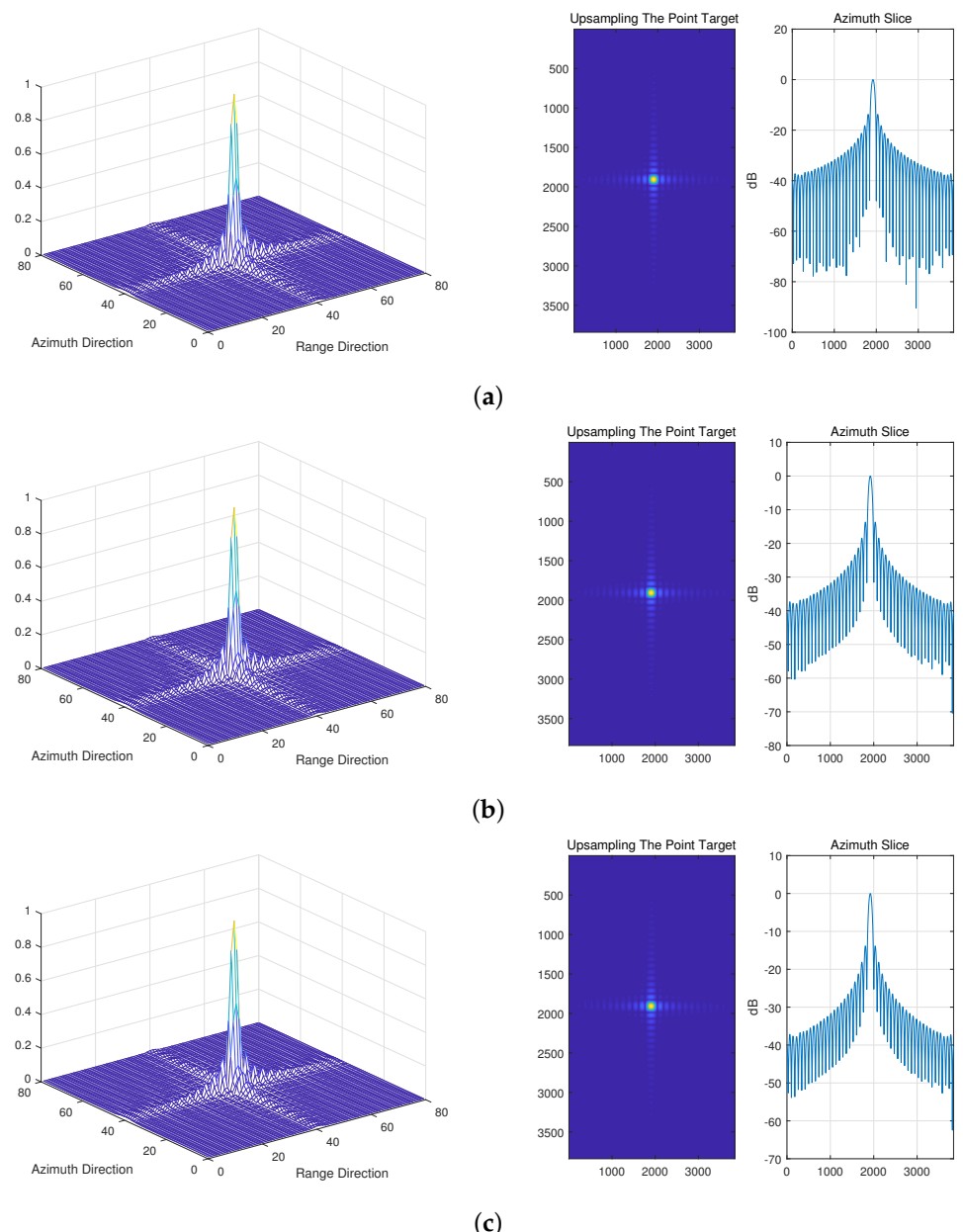

(a)

(b)

(c)

**Figure 8.** The imaging results of the parabolic antenna with different pointing deviations in the case of alignment. (**a**) Pointing deviation = 0°; (**b**) pointing deviation = 1°; (**c**) pointing deviation = 2°.

**Table 2.** The point target energy analysis in the case of alignment

| Pointing Deviation | $I_r$ | $I_s$ | $\Delta_s$ | $\Delta_t$ | $\lvert \Delta_s - \Delta_t \rvert$ |
|:---:|:---:|:---:|:---:|:---:|:---:|
| 0° | 12.67 dB | 12.95 dB | −0.28 dB | −0.28 dB | 0.00 dB |
| 0.5° | 12.63 dB | 12.82 dB | −0.19 dB | −0.20 dB | 0.01 dB |
| 1° | 12.49 dB | 12.69 dB | −0.20 dB | −0.20 dB | 0.00 dB |
| 1.5° | 12.27 dB | 12.48 dB | −0.21 dB | −0.22 dB | 0.01 dB |
| 2° | 11.97 dB | 12.27 dB | −0.30 dB | −0.31 dB | 0.01 dB |
| 2.5° | 11.59 dB | 11.69 dB | −0.10 dB | −0.10 dB | 0.00 dB |
| 3° | 11.16 dB | 11.06 dB | 0.10 dB | 0.10 dB | 0.00 dB |
| 3.5° | 10.67 dB | 10.47 dB | 0.21 dB | 0.21 dB | 0.00 dB |
| 4° | 10.14 dB | 9.78 dB | 0.36 dB | 0.36 dB | 0.00 dB |

### 4.1.2. Misalignment

This part mainly discusses the case shown in Section 3.2.1. The RCS pattern may be approximately a linear model when the pointing deviation is out of the 3 dB beamwidth. Similarly, the P-band imaging simulation is performed based on the RCS patterns with different pointing deviations, and the point target energy is extracted. The normalized RCS patterns are shown in Figure 9. Part of the imaging results is shown in Figure 10. The point target energy analysis is shown in Table 3.

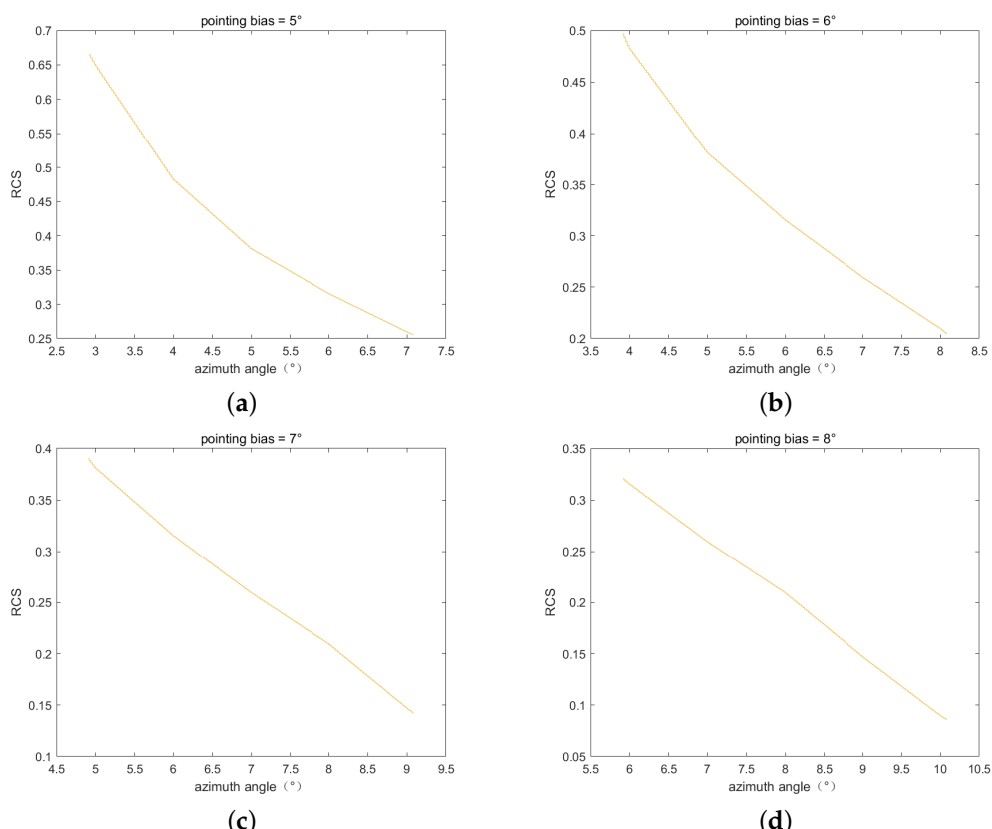

**Figure 9.** *Cont.*

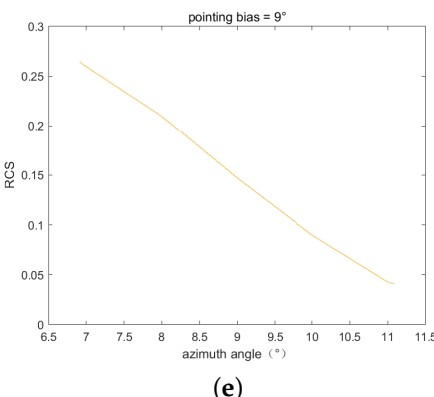

(**e**)

**Figure 9.** The normalized RCS pattern with different pointing deviations in the case of misalignment. (**a**) Pointing deviation = 5°; (**b**) pointing deviation = 6°; (**c**) pointing deviation = 7°; (**d**) pointing deviation = 8°; (**e**) pointing deviation = 9°.

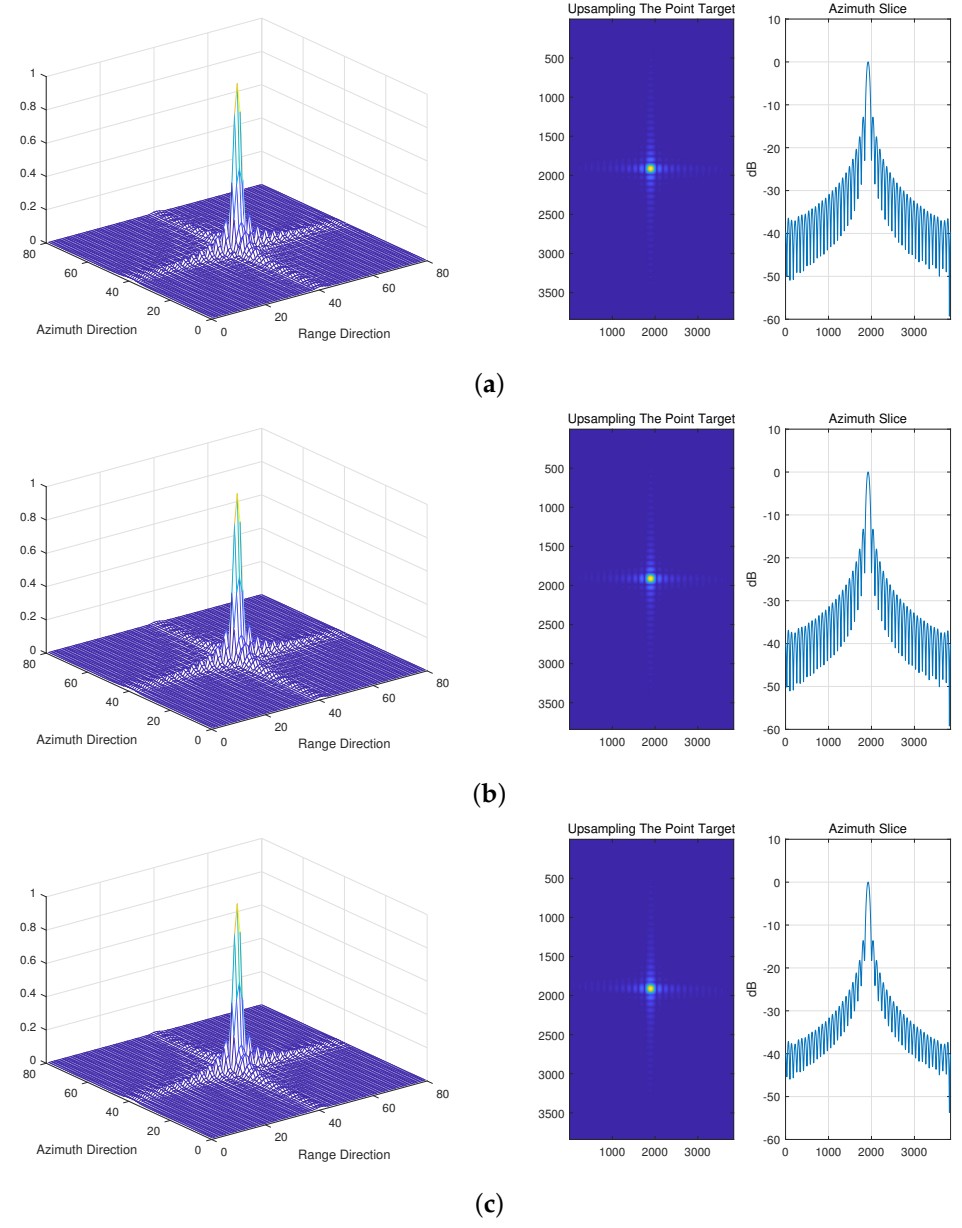

**Figure 10.** The imaging results of the parabolic antenna with different pointing deviations in the case of misalignment. (**a**) Pointing deviation = 5°; (**b**) pointing deviation = 7°; (**c**) pointing deviation = 9°.

**Table 3.** The point target energy analysis in the case of misalignment.

| Pointing Deviation | $I_r$ | $I_s$ | $\Delta_s$ | $\Delta_t$ | $\|\Delta_s - \Delta_t\|$ |
|:---:|:---:|:---:|:---:|:---:|:---:|
| 5° | 9.05 dB | 8.77 dB | 0.28 dB | 0.29 dB | 0.01 dB |
| 6° | 8.07 dB | 7.95 dB | 0.12 dB | 0.12 dB | 0.00 dB |
| 7° | 7.14 dB | 7.08 dB | 0.06 dB | 0.06 dB | 0.00 dB |
| 8° | 6.06 dB | 6.15 dB | −0.09 dB | −0.09 dB | 0.00 dB |
| 9° | 4.70 dB | 4.63 dB | 0.07 dB | 0.06 dB | 0.01 dB |

The results also verify the error model, and the difference between the theoretical error and the simulated error is less than 0.01 dB. In addition, as the RCS pattern gets closer to a linear model, the error introduced becomes smaller. When the pointing deviations are 7°, 8°, and 9°, the introduced error will be less than 0.1 dB, which verifies the conclusion of Section 3.2.1. Through this conclusion, the conditions of using the parabolic antenna have been relaxed. Even if the parabolic antenna is not fully aligned, some special angles make it possible for the parabolic antenna to be used.

*4.2. P-Band Absolute Radiometric Calibration*

After the target energy and RCS are known, the final radiometric coefficient $K$ can be evaluated according to the following formula:

$$K = \frac{I}{\sigma} \tag{28}$$

where $I$ is the point target energy, and $\sigma$ is the nominal RCS value of the calibrator. In the previous section, the point target energy at different observation angles is simulated so that the radiometric coefficient with different observation angles can be calculated. In order to verify whether the error model can improve the calibration accuracy, the error compensation is carried out as follows.

$$I_c = I_r - \Delta_t \tag{29}$$

$I_c$ is the point target energy after compensation, and the corrected radiometric coefficient $K_c$ can be obtained according to $I_c$. All the results are shown in Table 4.

**Table 4.** The calibration results after error compensation.

| Pointing Deviation | $I_r$ | $\sigma$ | $K$ | $I_c$ | $K_c$ |
|:---:|:---:|:---:|:---:|:---:|:---:|
| 0° | 12.67 dB | 45.71 dBsm | −33.04 | 12.96 dB | −32.75 |
| 1° | 12.49 dB | 45.45 dBsm | −32.96 | 12.70 dB | −32.75 |
| 2° | 11.97 dB | 45.02 dBsm | −33.05 | 12.28 dB | −32.74 |
| 3° | 11.16 dB | 43.84 dBsm | −32.68 | 11.07 dB | −32.77 |
| 4° | 10.14 dB | 42.55 dBsm | −32.41 | 9.78 dB | −32.77 |
| 5° | 9.05 dB | 41.52 dBsm | −32.47 | 8.76 dB | −32.76 |
| 6° | 8.07 dB | 40.70 dBsm | −32.63 | 7.95 dB | −32.75 |
| 7° | 7.14 dB | 39.85 dBsm | −32.71 | 7.08 dB | −32.77 |
| 8° | 6.06 dB | 38.92 dBsm | −32.86 | 6.16 dB | −32.76 |
| 9° | 4.69 dB | 37.38 dBsm | −32.69 | 4.63 dB | −32.75 |

Figure 11 shows the trend of the $K$ and $K_c$ for facilitating observation and comparison. The blue line is the radiometric coefficient $K$ calculated without compensation, and the orange line is the radiometric coefficient $K_c$ calculated after error compensation.

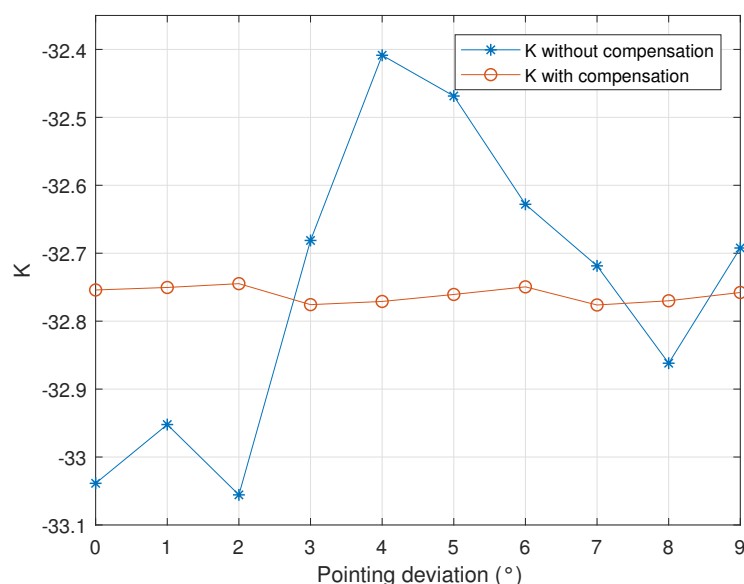

**Figure 11.** The comparison between $K$ and $K_c$.

The result shows that the variance of $K$ is 0.05, and the change of the pointing deviation will not cause the significant variance of $K$, which means that despite misalignment, the parabolic antenna can still be used for calibration. However, even if the pointing deviation ranges from the beamwidth, $K$ changes very drastically, and the difference between the minimum and maximum values reaches 0.65, which is an unacceptable error for the calibration. This verifies that, when the RCS pattern is a nonlinear model, it will introduce the additional error. If the calibration coefficients corresponding to the pointing deviations of 3°, 6°, 7°, and 8° are selected for comparison, the fluctuation is relatively small, and the RCS pattern at these points is closest to the linear model, which also proves the conclusion of Section 3: the closer the RCS pattern is to the linear model, the smaller the error introduced.

After the point target energy is compensated, the fluctuation of the radiometric coefficient $K_c$ is greatly reduced, the maximum difference is only 0.03 dB, and the variance is 0.0001. Compared with the situation before compensation, the performance is greatly improved, which is significant for improving calibration accuracy. The results prove the validity of the error model again, making it possible to use a parabolic antenna as a calibrator.

## 5. Discussion

The radiometric calibration of the P-band spaceborne SAR system faces the challenge of the lack of available passive artificial targets. Due to the large RCS and stable scattering characteristics, the parabolic antenna has the potential as a calibrator. It is generally believed that the parabolic antenna can be used for calibration only when it is aligned with the SAR sensor, but the narrow beamwidth of the RCS pattern and the wide SAR antenna beamwidth will introduce an additional error. Therefore, a quantitative error model that could be used for compensation should be established, and the conditions of using the parabolic antenna for calibration should also be reconsidered.

In this paper, the scattering characteristics of a typical parabolic antenna are simulated, and its azimuth pattern and the RCS over the range bandwidth are discussed separately. The results show that the effect of the frequency on the RCS of the parabolic antenna within 6 MHz bandwidth can be negligible, but the azimuth RCS pattern is sensitive to the azimuth angle. The influence of the azimuth RCS pattern on the calibration under the wide SAR beamwidth is discussed emphatically. The mathematical derivation shows that the inconstant azimuth RCS pattern will introduce calibration errors, and a quantitative formula of the error model is given. The conditions of using the parabolic antenna are analyzed based on the error model, combing the azimuth RCS pattern. The results show that the azimuth RCS pattern will introduce additional errors even when the parabolic antenna is aligned with the SAR sensor. When the azimuth RCS pattern is closer to a linear model, the smaller the error introduced. In order to verify this conclusion, imaging simulation is performed based on the parabolic antenna's RCS pattern with different pointing deviations. The difference between the error predicted by numerical simulation and the error predicted by the quantitative error model is less than 0.01 dB, which verifies the correctness of the error model. The results also prove that the closer the target's RCS pattern is to the linear model, the smaller the error introduced. Absolute radiometric calibration simulation is further applied, and the radiometric coefficient *K* with different observation angles is simulated. The results show that a non-negligible error will be produced when the parabolic antenna is aligned with the SAR. On the contrary, when the SAR sensor points out of the 3 dB beamwidth, if the RCS pattern is close to the linear model, the use of the parabolic antenna will not affect the final calibration result. After compensation with the error model, the calibration accuracy is improved efficiently. This research may increase the possibility of using the parabolic antenna as a calibrator for P-band SAR radiometric calibration.

## 6. Conclusions

This paper provides a discussion about using the parabolic antenna for P-band space-borne SAR radiometric calibration. Due to the fact that the narrow RCS pattern of the parabolic antenna will affect the calibration accuracy even when fully aligned, this paper establishes an error model for compensation. By using this error model, the conventional understanding that the traditional parabolic antenna can only be used when aligned is improved, which is meaningful for solving the lack of available reference targets for P-band spaceborne SAR radiometric calibration and for improving calibration accuracy. In the future, we will combine actual data for more verification.

**Author Contributions:** Conceptualization, J.H. (Jun Hong) and S.D.; methodology, Y.W. and S.D.; software, K.X. and S.D.; validation, T.Q. and S.D.; formal analysis, S.D.; investigation, J.H. (Jun Hong) and S.D.; resources, S.D.; data curation, S.D. and J.H. (Jianjun Huang); writing—original draft preparation, Y.W. and S.D.; writing—review and editing, Y.W. and S.D.; project administration, Y.W.; funding acquisition, Y.W. All authors have read and agreed to the published version of the manuscript.

**Funding:** This research was funded by the National Natural Science Foundation of China under grant number 61771453.

**Institutional Review Board Statement:** Not applicable.

**Informed Consent Statement:** Not applicable.

**Data Availability Statement:** The data presented in this study can be available upon request from the corresponding author.

**Acknowledgments:** The authors would like to express their gratitude to the anonymous reviewers and the associate editor for their constructive comments on the paper.

**Conflicts of Interest:** The authors declare no conflicts of interest.

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
