# Peer review of "Analysis of Using the Parabolic Antenna as the Passive Calibrator for P-Band Spaceborne SAR Radiometric Calibration"

_remotesensing, doi:10.3390/rs13214300_

Round 1

Reviewer 1 Report

In this paper, the authors propose the use of parabolic antenna for low-frequency SAR on behalf of traditional trihedral reflector (TCR). The proposal itself sounds somehow interesting however, the logic of this article is weak and the experiments are insufficient.

For example, the authors wrote that TCR is hardly applicable for P-band SAR for its weight, size, manufacture, and installation cost in L. 3. However, in this paper, the authors compared 9m TCR and 7.3m parabolic antenna, which are not so different in their size or manufacturing difficulty. Basically, the RCS is a function of the area which an antenna occupies and thus, the size of the antenna is somehow the same. In the radiometric SAR calibration, the requirement of the target is that it should be a point-like target. This is why TCR, a broad beamwidth reflector, has been widely used instead of parabolic antenna (as the authors mentioned). In such a context, if someone wants to use a parabolic antenna (or any other narrow beamwidth antenna) for the calibration, the one may rotate the antenna to keep reflectivity stable (e.g., https://doi.org/10.1155/2013/583865). Nowadays, we can forecast the satellite’s orbit precisely and therefore, we can face the antenna toward the satellite accurately.

Therefore, the configuration of the research seems impracticable because the authors fixed the antenna on ground and installed specific reflector in the focal area in order to discuss the linear assumption of the antenna pattern.

As written above, the answer to the authors’ question in L. 55, “whether the parabolic antenna can be used for P-band calibration still needs further discussion,” is clear. It can be used as long as we handle the parabolic antenna properly, and considering improper handling is unnecessary.

Of course, the authors can discuss for a fixed antenna case. However, in such a case, the radiometric calibration becomes unreliable because the antenna patterns of SAR and reflector are both unknown. The proposal deeply depends on the reflector’s antenna pattern in the practical use. It is quite unreasonable that such a large reflector’ antenna pattern can be estimated accurately. The merit of using TCR is its broad beamwidth which requires less accuracy (or just checking the angle of the triangles).

Author Response

Dear Editor, Dear reviewers

We’d like to express our most sincere gratitude to the reviewers for your effort and patience in reviewing our manuscript. We deeply appreciate your constructive comments that greatly help improve the technical quality and the presentation of this manuscript.

We have studied the reviewers’ comments carefully. The revised portion has been highlighted in our revised manuscript. Appended to this letter are our point-by-point responses to the comments and suggestions raised by the reviewers.

Thank you in advance for your time.

Sincerely,

Shaoyan Du

Reviewer 2 Report

This paper deals with a SAR calibration technique, more precisely a P-band SAR, using instead of a corner reflector, a dish antenna. An analytical analysis is done to analyze the behavior when subjected to broadband EM transmissions.
Figure 2 is not at all clear. You need to explain it better.
Figure 1 (b), explain more about the physical meaning of this figure and why it proceeds in an oscillating pattern?
Formula (5) is not understood at all. You need to explain it better. I suggest you start with the raw SAR signal, and perhaps give the analytical derivation of a backscattered SLC signal from a single target. It must be a double sync function.
I also don't see (6) as rigorous. The double sync, (the range sync and the azimuth sync) must be a function of both time and azimuth space, and must contain both the range and azimuth resolution parameters.
Explain (14) better. I don't understand (22) either. 
The experimental results are not understood. You need to explain it better by making a more comprehensive intoduction.
The project of the dish (or dishes) must be clearly visible, you must add the drawing with all the measures that you have dimensioned.

Author Response

(The authors gave the same response as above.)

Reviewer 3 Report

The paper presents new calibration approach, which might be of interest for P-band SAR calibration. But the content of the article raises a lot of questions.

  1. Line 39. Authors are not correct when writing “There’s a criterion on the calibrator that the calibrator needs to have sufficient RCS to ensure that it is visible in the SAR image[15]”. No, real criterion for the calibrator is signal to clutter ratio.
  2. Line 52. “The above experiments are performed in S- and C-band”. That is not correct. There were no experiments in S-band.
  3. Line 100 “Lunching” – should be “Launching”.
  4. Line 106: Commonly used RCS units are dBm2. So, it is incorrect to write 49.44 dB when speaking about RCS. That is typical error across the manuscript.
  5. Line 131: “sC is the average backscattering coefficient” – this definition causes some confusion. Commonly used name for sC is normalized radar cross-section.
  6. Line 152 “beamwidth of the TCR is about 20 deg”. No, it is about 40 deg.
  7. Fig 4a. Why there is RCS extremum at 433 MHz, if in theory there is monotonous RCS dependence from frequency?
  8. In eq (14) symbol K is used to name processing gain and Doppler frequency rate at the same time.
  9. Integration of sigmaL from (23) is done in (24) incorrectly
  10. Integration of sigmaQ from (25) is done in (26) incorrectly
  11. Exponential term ”eta” in (27) should be divided by 10
  12. Plots in Figs.7 and 9 represent not RCS, but normalized RCS.
  13. An importance of Fig.8 is unclear. The figure is not commented in the paper.
  14. Taking in mind severe mistakes in (24) and (26), the numerical results of the article in section “Experiments” seem to be doubtful.
  15. If we use misaligned parabolic antenna for the calibration, its backscatter may be reduced significantly. According to the paper, the RCS loss will be, for example, about 5 dB at 6 degrees angle. Because of that the signal to clutter ratio may be too low for the calibration purposes... I mean to say, that antenna depointing is unwelcome.
  16. The paper should be revised thoroughly.

Author Response

(The authors gave the same response as above.)

Round 2

Reviewer 1 Report

The revised manuscript contains several problems.

Problem statement is still not clear especially for the restriction of using parabolic antennas instead of TCRs.

Some sentences use interrogative sentence (i.e., L. 76).

Please reconsider the effective digits. In general, MATLAB shows the 4th order of the value as a default setting while it doesn’t ensure the validity.

Author Response

Dear Editors, Dear reviewers

We’d like to express our most sincere gratitude to the reviewers for your effort and patience in reviewing our manuscript. We deeply appreciate your constructive comments that greatly help improve the technical quality and the presentation of this manuscript.

We have studied the reviewers’ comments carefully. The revised portion has been highlighted in our revised manuscript. Appended to this letter are our point-by-point responses to the comments and suggestions raised by the reviewers.

Thank you in advance for your time.

Sincerely,

Shaoyan Du

Reviewer 2 Report

Accepted

Author Response

Dear Editor, Dear reviewers

We’d like to express our most sincere gratitude to the reviewers for your effort and patience in reviewing our manuscript.

Sincerely,

Shaoyan Du

Reviewer 3 Report

Yes, authors did their best when working out the reviewer remarks. The paper quality has improved significantly.

But in some cases they overdid with corrections. And the reason is that they use the term “backscattering coefficient” when in some cases it is appropriate to write “radar cross section, RCS” or sometimes “normalized radar cross section, NRCS”. The units for RCS are dBsm, and for NRCS – just dB. Authors changed correct version of the text and wrote in lines 141-144 mistakenly:

“In the  P-band, the scattering coefficient of the desert is about -40 dBsm, the scattering coefficient of the cultivated land is between -38 dBsm and -30 dBsm, and the scattering coefficient  of the forest is about -23 dBsm. Without loss of generality, assuming the background scattering coefficient sC = ?25dBsm,....”

The mistake is that for the distributed targets people usually use to use NRCS values. And according to the numbers in revised paper, those are normalized radar cross sections, in decibels, not decibels of square meters. Consequently, for example, not “scattering coefficient of the desert is about -40 dBsm”, but “scattering coefficient of the desert is about -40 dB”.

Author Response

(The authors gave the same response as above.)
